# Sponge City and Water Environment Planning and Construction in Jibu District in Changde City

**Yumei Deng, Jie Deng and Chun Zhang \***

School of Municipal and Mapping Engineering, Hunan City University, Yiyang 413000, China
\* Correspondence: zhangchun7912@163.com

**Abstract:** Urban waterlogging and urban water environment problems in Changde city caused by extreme weather have seriously hindered the sustainable development of cities. A sponge city not only involves the inheritance and development of foreign technology but also a new method for its use. The background of sponge city construction based on green infrastructures in China was introduced in this study. As one of the first pilot construction cities based on the sponge concept, Changde city possesses natural geographical advantages. The current urban situation, rainfall type and water environment in the sponge construction area were analyzed and the causes of urban waterlogging and deterioration of urban inland river water quality are presented. Based on the urban water environment and ecological status, the specific strategic objectives of the sponge city transformation are given. Meanwhile, the overall technical route and the concrete realization path of each index, such as the water environmental system, water ecological system and security system, are also presented. The annual net flow total control rate and the runoff pollution reduction reached 77.56% and 45.18%, respectively. The total runoff and peak flow were also reduced by 35.08% and 26.82%, respectively. Meanwhile, the peak flow of runoff pollution concentration was reduced by 31.99%. The pollutant load reduction rate of non-point source pollution in the area reached more than 45%. The project not only alleviated the problems of urban waterlogging and black and odorous water bodies but also ensured the sustainable development of the urban water environment.

**Keywords:** sponge city; waterlogging; water environment; sustainable development

## 1. Introduction

At the conference of Low-Carbon Cities and Regional Development Technology Forum held in April 2012, the sponge city concept was first proposed by the Ministry of Housing and Urban-Rural Development of China [1]. In addition, the technical guide for sponge city construction, 'Construction of Rainwater System for Low Impact Development (Trial)', was promulgated and implemented on 22 October 2014, which provided guidance and the basis for the construction of sponge cities [2,3]. A sponge city is a new generation of urban stormwater management concepts that gives cities the resilience required to adapt to environmental changes and deal with natural disasters caused by rain [4]. In recent years, the problems of waterlogging in Chinese cities have become more serious due to the rapid development of urbanization and the rapid change in the global climate. These problems not only threaten urban production and life but seriously affect city development. Given this, solving the problem of urban waterlogging is of great significance. However, traditional waterlogging prevention was usually completed via a pipeline water transfer and urban rainwater recycling was ignored. Furthermore, the traditional methods have deficiencies, such as high cost, short life and insufficient drainage standard. Therefore, sponge city planning and construction have profound significance for solving the problem of urban waterlogging [5,6].

As a new type of urban rain flood management concept, it actually refers to a city like a sponge and has satisfactory elasticity for adapting to environmental changes and

responding to natural disasters. The rain will be impounded and treated when it rains and be released and reused when needed. The main design concept of a sponge city is to realize the natural accumulation, penetration and purification of rain in an urban region to the maximum under the premise of ensuring the city's flood control and drainage safety and to promote the utilization of rainwater resources and the protection and restoration of the ecological environment. Meanwhile, lower maintenance and landscape sustainability should be employed to avoid high future operating costs and to realize better social and ecological benefits [7]. The sponge city design and construction includes four principles, which are penetration, detention and storage regulation, purification and utilization, and discharge. In fact, sponge cities are common in our life, such as green space, a pond or a flower bed. They all have the characteristics of a sponge city. The difference between them and our current sponge city design lies in the efficiency of sponging. Of course, there are various ways to express the sponge city concept; different types and forms of sponge facilities can be made according to different conditions and implementation methods [8]. In order to accommodate different sites and meet their functions, different forms of expression or means of realization are used, such as bio-retention ponds, rainwater gardens, low elevation greenbelts, high rainwater flower beds, plant ditches and underground reservoirs. Generally, a sponge city design is based on the site conditions to actualize better water ecology, ensure water safety, improve the water environment and provide optimal utilization of water resources [9]. The sponge city system diagram is presented in Figure 1.

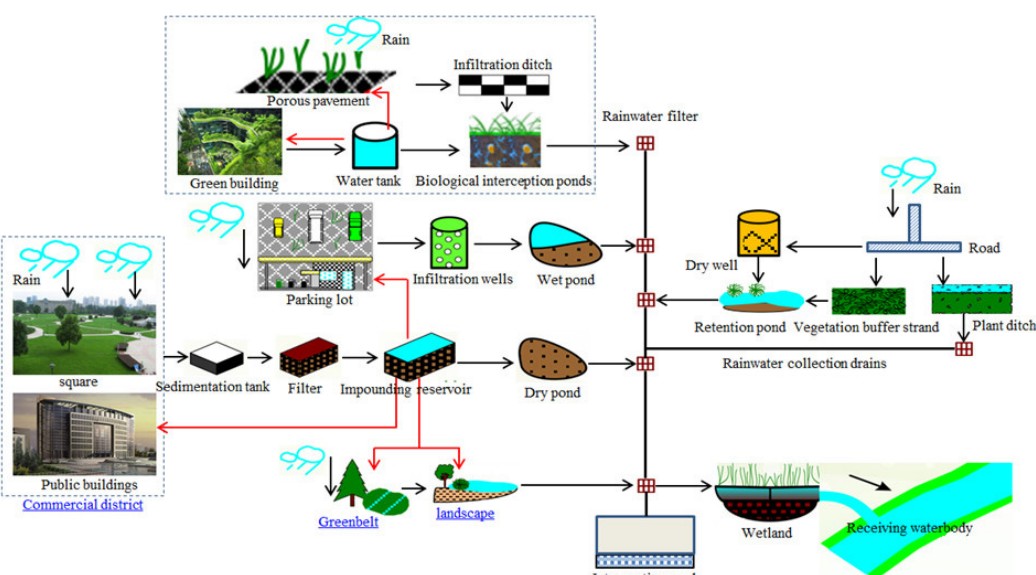

**Figure 1.** The diagram of a sponge city system.

Based on a series of policymaking and related documents, the first pilot cities of sponge construction, such as Wuhan, Jinan, Changde and Pingxiang, were announced in China in 2015. As one of the first pilot cities of sponge construction, Changde city has its specific advantages for the programming and construction of a sponge city. Changde is adjacent to Dongting Lake and is a typical water-sensitive city with a developed water system; the average annual precipitation is 1200 mm. Considering the special planning of a sponge city in Changde, the natural environment, the planning concept of development level and the economic development of the area, among other conditions, the total annual runoff control rate in this planning region in Changde was determined to be 78%, and the corresponding designed rainfall was 21 mm. According to the goal of water quality based on the 'Water Environment Function Zoning of Main Surface Water System in Hunan Province (DB43/023-2005) and the 'Special planning of Sponge City in Changde' (2015~2030), the water quality target of all water bodies in the area was class IV. The suspended solid (SS) index is used as the runoff pollutant control index and was determined

to be 45%. The discharge standards of rainwater drains should reach a recurrence interval of 2 years, and the major roads in important areas and general areas should reach a recurrence interval of 3~5 years. The flood control standards of once every 100 years were adopted. All of the relative technology parameters were determined according to the layout scale and the construction requirements of the related facilities in the sponge city in this area [10,11].

The other parts of the paper are organized as follows: Section 2 reviews the extant literature on the application of the advanced urban rainwater utilization system in different countries. Section 3 highlights the methodology and data used in the research. It presents the theoretical framework and the empirical model used in this attempt. The results and discussion are presented in Section 4. It presents the planning and design process of this sponge city and its operation, as well as the improvement of the urban water environment of the city. The conclusions are provided in Section 5. As one of the first sponge city pilot construction cities in China, it can provide a reference for later sponge city planning and design in other cities.

## 2. Literature Review

In the 1980s, low influence development (LID) technology was proposed based on the rain garden in Prince George County in America. Subsequently, the first comprehensive design technical standard was compiled in 1999 by the United States Government. The design concept of LID was approved rapidly by the United States Federal Government and the state governments between 2000 and 2005 [12,13]. In the following five years, LID was applied to rain management and the design of the control of non-point source pollution by the US Environmental Protection Agency [11,14]. Presently, such a 'Green Infrastructure' concept is promoted in the United States [15–17].

In addition, the sustainable drainage system, abbreviated SuDS, is an integral part of the BGI approach to water management; it is aimed at making urban drainage systems compatible with the natural water cycle [18]. SuDS may consist of 'natural' measures, such as green roofs, planters or green belts, combined with 'artificial' measures, such as underground infiltration and retention tanks [19]. The blue-green infrastructure approach can perhaps best be summarized as building with nature, as opposed to only reacting to it, to solve urban challenges. It aims to secure a sustainable future while also generating multiple benefits in the environmental, ecological, social and cultural spheres. Moreover, it requires a coordinated interdisciplinary approach to water resource and green space management from institutional organizations, industry and academia, as well as local communities and stakeholders [20,21]. In recent years, conventional 'hard engineering' approaches (sewers, etc.) have been complemented—and in some cases replaced—with more natural approaches using blue–green infrastructure. This can involve, for example, a system of measures such as green belts, grassed dry retention ponds or rain gardens. Integrated planning and modeling have also become a key part of stormwater management in urban areas. SuDS are sometimes incorrectly viewed as being more expensive than conventional stormwater management. This misconception comes from too narrow a focus on investment costs for projects in city centers. In contrast, they are usually more cost-effective than the conventional methods and they can also bring additional multiple benefits. Experiences from the United States can give us some useful perspectives in this respect when considering the financial costs of green infrastructure in general. Engineering cases from many European countries also suggest that the maintenance costs and responsibility for the maintenance of sustainable drainage systems (SuDS) are usually more challenging than the cost of implementing SuDS in new developments. Consequently, arrangements for the maintenance of SuDS systems should be considered during the early stages of design; otherwise, there is a risk of turning a good SuDS implementation into a non-functional set of facilities [21,22].

Water-sensitive urban design (WSUD) in Australia emerged on account of a similar concept. A SuDS imitates the natural processes using the slow release of the stored

rainwater, promotion of rainfall infiltration and pollutant filtration [23]. As an innovative multi-field comprehensive discipline developed in the research and practice of water management in Australia over the past two decades, water-sensitive urban design is gaining more and more recognition in the international scope. In the continuous research and summary on the evolution of the urban water industry with human development, Australia has proposed the evolution process and future of urban water [24]. The roadmap of each stage is presented in Figure 2 and the final vision and imagination of a water-sensitive city were proposed.

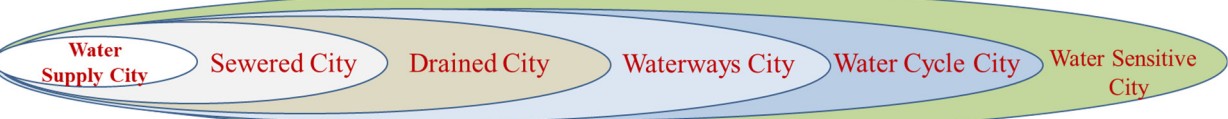

**Figure 2.** The roadmap of each step of urban water in Australia.

A vast majority of cities in developed countries have gone through multiple stages from a water supply city to a water cycle city. The LID in America, SuDS in Britain and WSUD in Australia have experienced long-term development at different stages, where the proposal of low-impact development focuses more on the control of small- and medium-rainfall events and takes the total runoff as the control target, and adopts the measures of source, dispersion and small green technology. However, the future development of Chinese cities does not necessarily need to follow the old roadmap and start from the drainage city again to reach a future sponge city through the waterways city and water cycle city. Most cities in China are now in the primary stage of being waterways cities; urban development is still active and a large number of regional rectification and development projects still are in the planning stage [25,26]. Sponge city construction aims to transform the concept of urban development and build a sustainable urban development model. Its core concept is consistent with American low-impact development. However, the low-impact development technology measures in sponge city construction include not only the green facilities at the source but also the green rainwater infrastructure at different links and scales in the middle and end. Control targets should include comprehensive targets, such as the total runoff, peak value, pollution and frequency. Although the original intention of sponge city construction is related to the solution of urban waterlogging, it should return to the premise of how to improve the sustainability of urban construction and urban livability in the final analysis, which also coincides with the connotation of a water-sensitive city.

## 3. Methodology

### 3.1. Analysis of the Current Situation of the Planning Area

The total planning area was 10.3 km$^2$. The underlying surface analysis was carried out based on the 1:2000 topographic map and was divided into five categories: buildings, greenbelts, water system, pavement and roads. According to the analysis, the overall green rate of the area was high and the current underlying surface mainly consisted of pavement, buildings, greenbelts and roads that occupied 31.34%, 23.68%, 24.85% and 14.39% of the area, respectively. The analytical diagram and the statistical table are presented in Table 1.

**Table 1.** The statistics of the underlying surface in the planning area.

| Category | Area (ha) | Proportion (%) |
|----------|-----------|----------------|
| Building | 243.90 | 23.68 |
| Greenbelts | 255.96 | 24.85 |
| Water system | 59.12 | 5.74 |
| Pavement | 322.82 | 31.34 |
| Road | 148.20 | 14.39 |
| Total | 1030.00 | 100 |

The area was an alluvial plain along the Yuanjiang River that was relatively flat on the whole. The elevation was mainly in the 31~34 m range and the average elevation was 32.6 m. The topography changed little; the slope was generally higher in the north and lower in the south. The slope is basically less than 2 degrees; the average longitudinal slope was 0.5% and the fluctuation was basically less than 0.9 m. The maximum depth of the groundwater at each survey point was less than 1.5 m, belonging to the shallow groundwater buried area. The deepest groundwater level was less than 6 m and was greatly affected by the water level of the Yuanjiang River. The elevation and slope distribution in the planning area are presented in Figure 3.

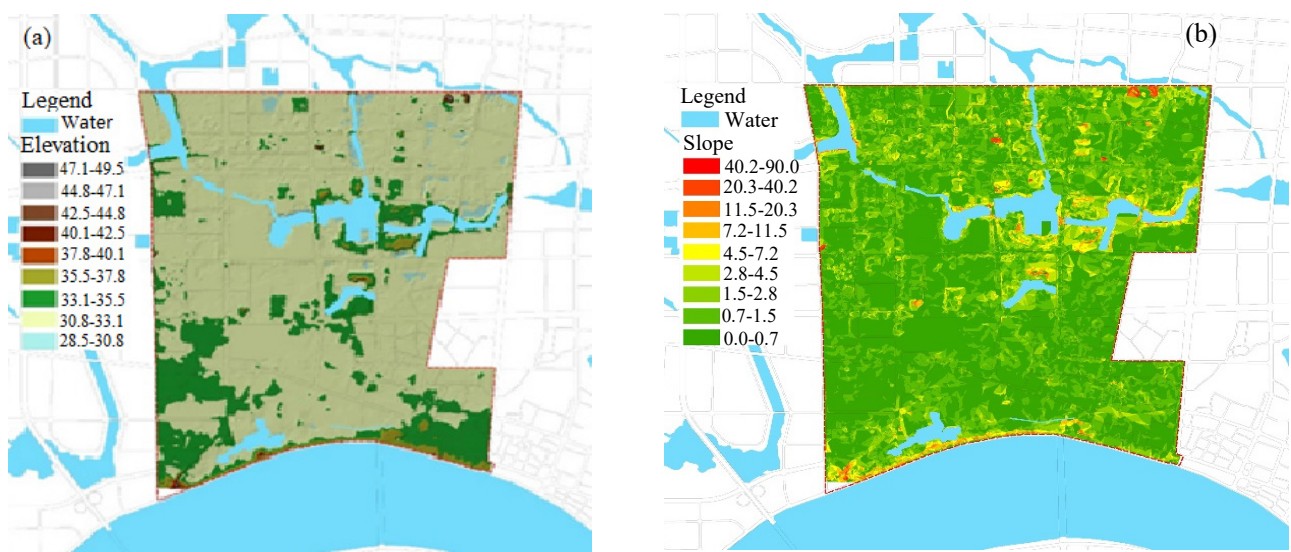

**Figure 3.** The elevation (**a**) and legend distribution (**b**) in the planning area.

*3.2. Analysis of the Rainfall*

3.2.1. Short-Duration Rainfall Pattern

The adopted rainstorm intensity formula was as follows:

$$q = 167i = \frac{1146 + 916.5\lg T}{(t + 5.4094)^{0.5692}}$$

Twenty-five maximum hourly rainfall values were determined according to thirty-four years (1987~2020) of continuous minute-level rainfall data to determine the short-duration rain peak location. The results showed that the rainfall pattern was mainly unimodal rain. Furthermore, the rain peaks mostly occurred in the front and middle parts. Uniform and bimodal rain patterns were less and the rain peak mostly occurred between 0.21 and 0.53. Ultimately, the short-duration rain peak location in Changde was determined at 0.4 based on the comprehensive statistics. Then, the accumulated rainfall in each time interval of the rain could be calculated. The short-duration rainfall intensities with recurrence intervals of three and five years were presented in Figure 4.

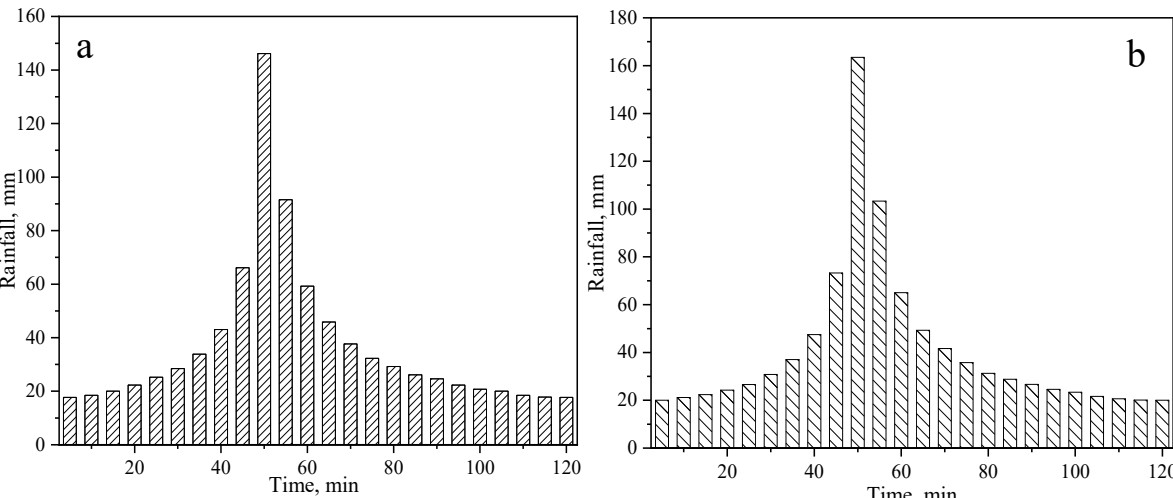

**Figure 4.** The short-duration rainfall intensities with recurrence intervals of three (**a**) and five years (**b**).

### 3.2.2. Long-Duration Rainfall Pattern

In order to determine the standards for waterlogging prevention and control, the long-duration rainfall pattern should be studied during the planning stage. The twenty-four-hour duration of a rainstorm was adopted when designing the project of urban drainage and waterlogging prevention works to evaluate the waterlogging capacity. One hour was determined as the time frame of the designed rain pattern and the total twenty-four-hour rain duration was determined. According to the twenty-four-hour precipitation process of the heaviest twenty rainstorms in the past thirty-four years in the statistical sample, the twenty-four-hour-long rainstorm pattern was preliminarily obtained. The urban waterlogging control standard of Changde city was thirty years and the corresponding total twenty-hour rainfall was 206.59 mm. The maximum hourly rainfall was 79.06 mm and the maximum rainfall for three hours was 117.67 mm. The rain peak approximatively occurred at the fourteenth hour. The twenty-four-hour rainfall distribution with a recurrence interval of 30 years is presented in Figure 5.

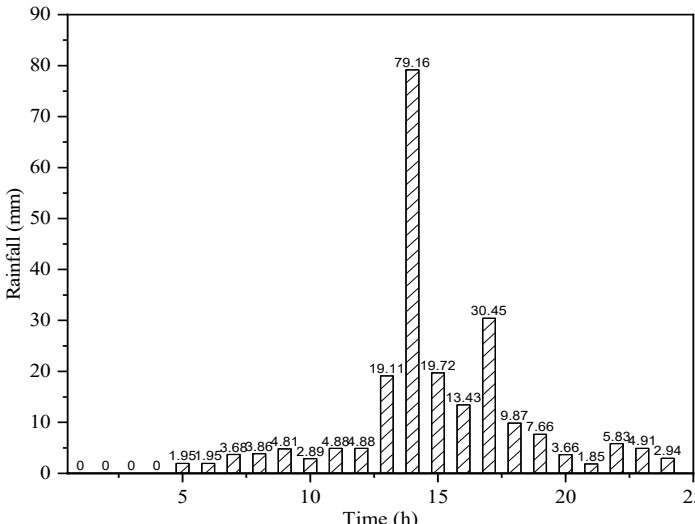

**Figure 5.** The 24 h time distribution chart of the rainfall with a recurrence interval of thirty years.

### 3.3. *Analysis of the Present Water Environment*

The quality of the water environment in Changde is stable on the whole according to the evaluation of the monitoring indicators. The current situation of water environmental

quality in the planning area is presented in Figure 6. The five exploratory points ( ☆1#, ☆2#, ☆3#, ☆4#, ☆5#) are the position for measurement of groundwater level. The water in the Xinhe Canal, Whitehorse Lake and Chuanzi River belonged to water quality criteria band IV. The water quality of the moat river in the old town was seriously polluted due to the overflow of a large amount of confluence rainwater in the city and it was classified as being in category V or inferior V. The flood control works in this planning area were the Yuanshui levees and the flood control standard was one hundred years. The sole drainage pumping station in this area is the ship terminal pumping station that was remolded in June 2014. The remolded pumping station can enhance the ratio of the wastewater transported to the sewage plant and reduce the quantity and concentration of the wastewater emitted directly. The combined sewage could be cleared through the water carrying ecological filter to provide clean water for the Chuanzi River. The available volume of the detention ponding should be ensured and the river should be dredged so that there is no surface runoff from a rainstorm that would cause flooding. Furthermore, the drainability of the rainfall pumping station was 12.6 $m^3$/s and 500 L/s for the rainfall and non-rainfall periods, respectively. Moreover, the project also included the enclosed sedimentation tank (1a + 1b) with a total volume of 7000 $m^3$. The sizes of the detention ponding (2#) and the water-carrying ecological filter were 13000 $m^3$ and 8400 $m^3$, respectively. As the most important water treatment facility in the reconstruction of the ship terminal, the built ecological filter was combined with the river landscape, which is shown in Figure 6. The rainfall in the inferior V category can be treated to reach category IV. The cleared rainfall can be provided to the Chuanzi River as the landscape and supplementary water to enhance the water quality.

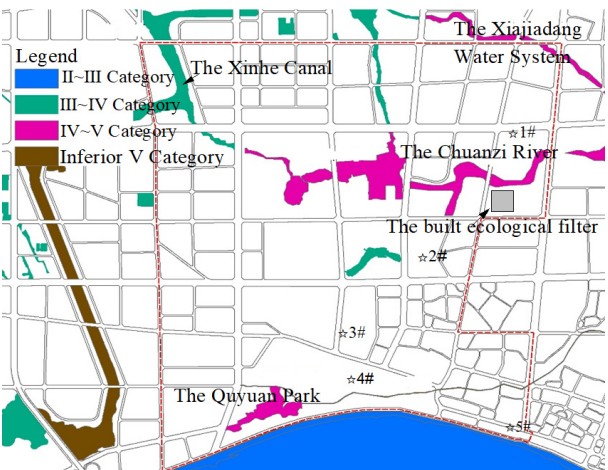 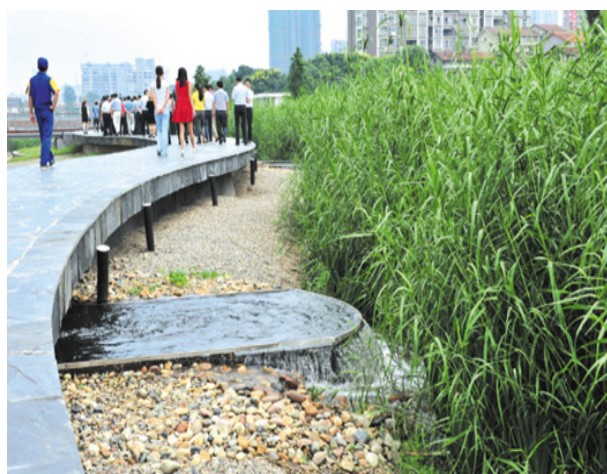

**Figure 6.** Current situation of water environmental quality and the built ecological filter in the planning area.

### 3.4. Analysis of Current Construction Conditions

The land in the planning area was mainly used for residential and public administration and service facilities, which accounted for 45.5 and 14.5 percent, respectively. Based on the actual situation of the planning area, six ecological factors were selected as the analytical basis: topography, geology, soil, hydrology, bio-diversity and land-use type. The area of the suitable ecological protection area, comparative suitable ecological protection area and the general suitable ecological protection area were 0.55, 0.83 and 8.95 $km^2$ and occupied 5.21%, 7.96% and 86.83%, respectively. The year 2017 was determined as the typical year because the rainfall approached the average rainfall calculated using several decades of data. According to the simulation results of the model, the current annual total runoff control rate of the area was 46.3% and the overall hardening rate of the area was high. Therefore, it was necessary to put forward the index requirements for the built-up

area and make synchronous improvements in the next step of the renovation process to further reduce the urban surface runoff coefficient. Changde was identified as one of the first batches of pilot sponge cities in China in 2015. Preliminary progress was made in the sponge city construction and only one water point is left on Changgeng Road in this planning area. The distribution of the waterlogging water points in the planning area is shown in Figure 7. The position 1 was the only remaining waterlogging water point and the other five points have been reconstructed. In the case of waterlogging, the catchment area was about 2.56 ha, the retention time was about 10 h and the water depth was about 30 cm in this region. The main reason leading to the waterlogging was the partial reverse slope of the main trunk pipe and the relatively low terrain, which resulted in a large amount of rainwater being concentrated and a hydraulic bottleneck.

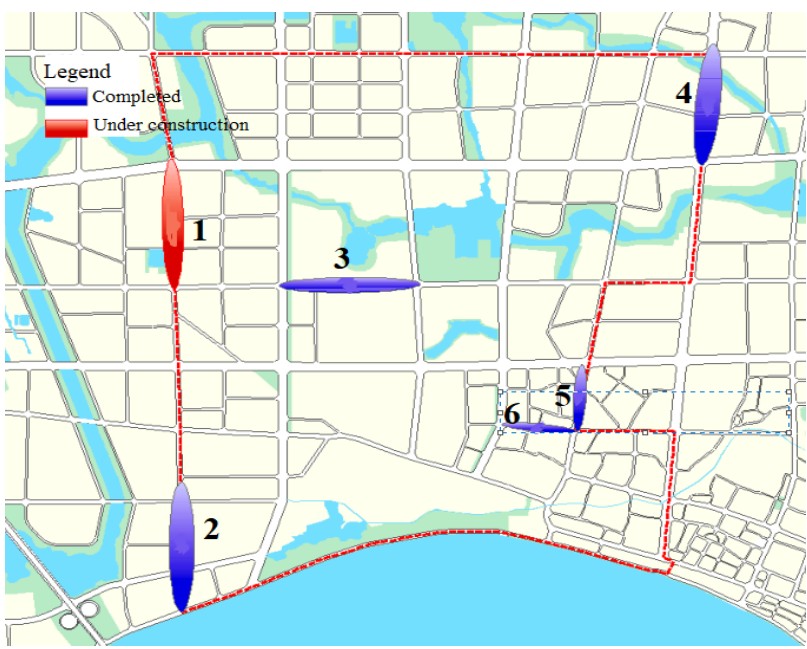

**Figure 7.** The distribution of the waterlogging water points in the planning area.

### 3.5. Main Problems and Cause Analysis

The water body in the planning area was seriously polluted. The drainage system of the moat in the old city in the planning area was the intercepting combined confluence. It was constructed in the 1960s and 1970s. With the continuous development of the city, the urban area was expanding. The intercept multiples are currently between 0.5 and 1.5. According to the relative area of Changde and the special planning of the surrounding river system, the water body of the moat was polluted by the overflow of the sewage from the combined sewerage system when the rain recurrence interval was one year. Simultaneously, due to the high groundwater level, the groundwater flowed into the pipe network in the flood season, which resulted in an increase in the overflow times of the combined drainage system.

The water ecosystem in the planning area was weakened. The control rate of the total annual runoff was low. Most of the drainage system around the moat in the planning area was a separate drainage system. However, there were a lot of problems, such as a hybrid junction and leakage of the pipe network. Furthermore, due to the high groundwater level, a large amount of groundwater was seeping into the sewage pipe network, resulting in the low inflow concentration of the sewage plant and the increase in sewage volume. According to the confluence model built previously, the annual total runoff control rate of the current area was found to be 46.33%, which showed that the overall hardening rate and the runoff coefficient of the current area were high. The overall annual total runoff

control rate of the current area was far below the requirement of a 78% control rate. The distribution of the runoff control rate is presented in Figure 8.

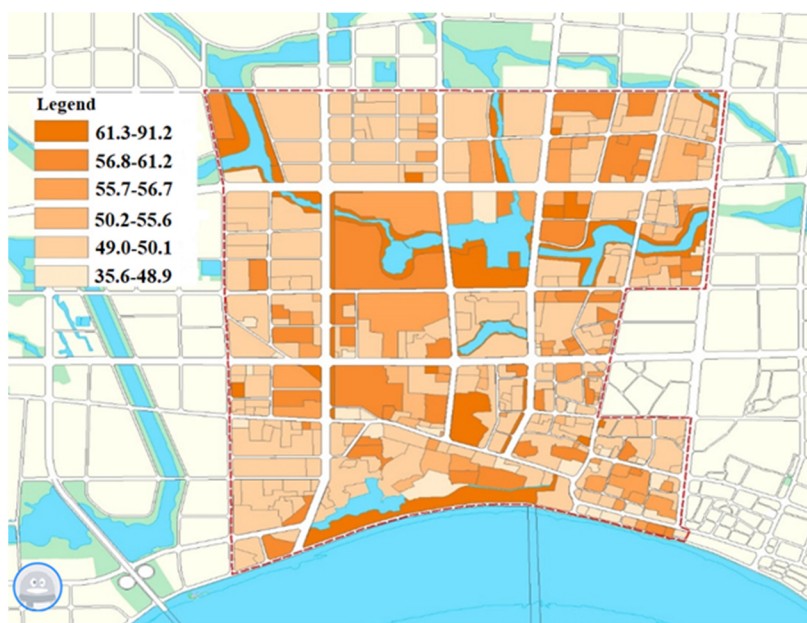

**Figure 8.** The distribution of the runoff control rate of the current area.

Additionally, the river channels were hardened and the ecological construction was previously ignored. Houses were built on the moat in the Jiangbei urban area, which made part of the moat into a dark ditch. About 35% of the river courses in this area have problems with river hardening and channelization. At the same time, there is basically no runoff in the river courses in the area. The ecosystem is fragile, the biodiversity is low and the natural ecology is not reflected.

The water use efficiencies were universally low, and the waste of water is serious in the planning area. Presently, the main water used in the area, such as domestic water, non-domestic water and water for special industries, are all supplied by waterworks, except for a small amount of rainwater used for landscaping from rivers and lakes. Moreover, the rainwater utilization in the area was mainly scattered and distributed. The rainwater utilization facilities in Hunan University of Arts and Sciences, Whitehorse Lake Park, Dingquyuan Park and part of the courtyards provided water for urban and residential landscaping, sanitation, landscape and so on.

The annual water demand of the road and green space in this area was 1225 million m$^3$, and the annual rainwater collection capacity of this area was 3923 million m$^3$, which could fully meet the water demand. However, the rainwater collection and utilization facilities were not matched then and the utilization rate was low. Therefore, the utilization rate of non-traditional water resources needed to be increased.

## 4. Results and Discussion

### 4.1. Objective and Strategy

According to the concept of 'water saving priority, spatial balance and systematic governance', a sponge city construction framework that conformed to the natural environment characteristics of the airport area and the actual urban development was put forward. The planning ideas aimed to lead the urban development with the sponge city construction concept; to promote ecological protection, economic and social development, and cultural inheritance; and to build a new image of Changde with ecological, safe and dynamic sponge construction, and then to realize the development strategy of 'water security guarantee, water environment improvement, good water ecology, beautiful water landscape'. This

study aimed to solve the urban waterlogging and urban water environment problems in the city and provided empirical reference of the planning and construction of the sponge city for other cities with the personalized design case.

In order to realize the above objective and to solve the problem of water pollution in the area, the point source pollution and runoff pollution must be reduced to make the annual production rate of COD, SS, TP and NH$_3$-N reach the aims of each district. Simultaneously, the target of the annual total runoff control rate of 75% should be satisfied by improving the control of the runoff of rainwater and solving the waterlogging problem in the area and by using comprehensive treatments at the source, during the process and at the end. Moreover, the recycling of rainwater in the origin and ending processes should be emphasized to solve the low utilization rate of rainwater resources.

### 4.2. Water Environment System

4.2.1. Overall Technical Route

The technical route taken for the overall scheme design was divided into four parts, which were, in turn, the overall goal, the overall scheme design ideas, the area construction scheme and the safeguard measures. First, according to the sponge city special and the overall planning of the city, the strategic objectives of the sponge construction of Changde were proposed. In addition, the annual runoff total control rate and the total removal rate of the annual SS could also be determined. Second, the overall scheme design idea of the area was determined, and solutions to problems such as water security, water environment and water ecology were proposed using source reduction, process control and end treatment, respectively. Third, the current situation elements of the planning area were evaluated and identified and the specific construction plan of the area was formulated on the basis of the current situation investigation, data combing and integration. Moreover, the drainage model was constructed to analyze the specific causes of the weak water ecology of waterlogging and water pollution in the area. The sponge system of water security, water environment and water ecology was constructed to determine the engineering quantity of the layout of sponge buildings, roads and squares, parks and green spaces, rivers and lakes, and related infrastructure. The sponge city planning model of the area was constructed to evaluate the compliance of the reconstruction of the area. Finally, the guarantee measures after the indicator landing and the project implementation were put forward, including the guarantee system, monitoring and assessment system, technical standard system and other guarantee measures.

4.2.2. Water Environment System

The construction of a sponge city has high requirements for water environment management. The water quality standard of surface water should be better than class IV, and the annual SS removal rate should be over 45%. Based on the above requirements, specific control strategies for point source and non-point source pollution were put forward by improving the sewerage pipe network, accelerating the construction of sewerage treatment plants and eliminating direct sewage discharge. Therefore, the pollutants could be reduced in the planning area and the pressure of a surface water environment also could be relieved. The technical remediation route of the water environment is presented in Figure 9.

The main water systems in the planning area include Chuangzi River, Xinhe Canal, Whitehorse Lake and Quyuan Park. The surface water quality standards of each water system should be defined according to the 'specialized planning of Sponge City in Changde'. Simultaneously, other water systems not mentioned in the zoning were calculated according to the class IV standard of water environment management for a sponge city. The surface water capacity was calculated using the fully mixed model presented as follows:

$$W = W_{dilution} + W_{self-purification} \tag{1}$$

Since the current water quality of the river system in the planning area was the same as the target water quality, the current dilution capacity was considered to be zero and the self-purification capacity of the river was the environmental capacity of the river:

$$W_{self-purification} = k \cdot V \cdot C_s \tag{2}$$

where $W_{self-purification}$: degradation capacity, $t \cdot a^{-1}$; K: integrated degradation coefficient, $1/d$; V: water volume, $m^3$; Cs: water environmental quality objectives, mg/L.

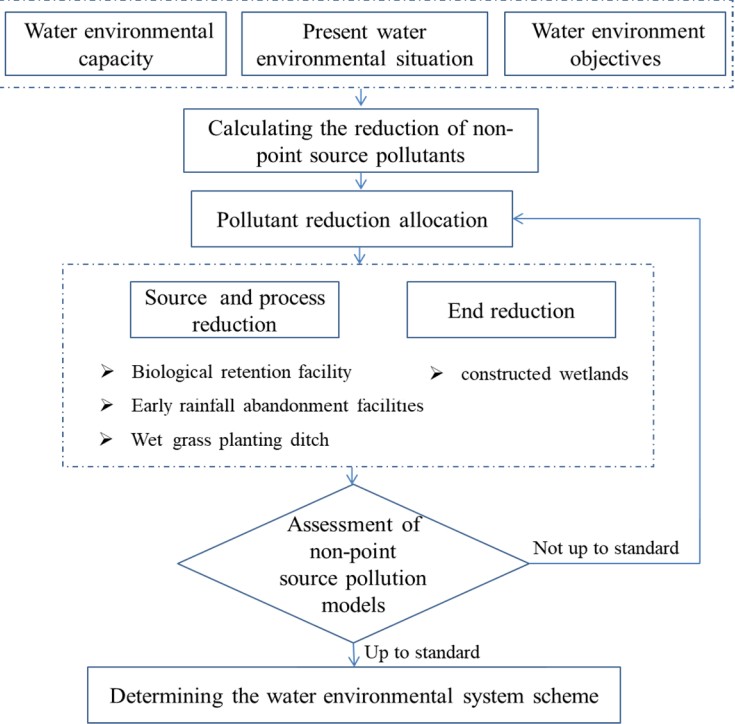

**Figure 9.** Technical remediation route taken for the water environment.

The point source pollutants in the planning area could be predicted using the method of emission coefficient according to the planned population. In addition, according to the analysis results of the underlying surface and the water quality testing data, along with the non-point-source pollution in the simulated area based on the SWMM software, the emissions could be further calculated.

On the basis of the reduction rate of SS of the long-term target, the amount of pollutants that needed to be reduced was calculated and the scale of source measures could be determined accordingly. Nevertheless, whether the source measures can be implemented should be deliberated while considering the actual plot condition. If not, the scale of source measures should be lowered and the measures should be ascertained. After removing the remaining water environmental capacity, the end reduction measures were calculated and it was determined whether all the end reduction measures could be implemented based on the actual plot situation. If not, the SS target reduction rate should be adjusted until all the measures could be implemented. The technical route taken is presented in Figure 10.

The point source pollution was controlled using the following methods. The sewage and rainwater pipelines in the residence community and the confluences of the municipal pipelines were remolded. Moreover, the ecological filters constructed in the ship terminal area could also reduce the point source pollution. The sewage purification center in Changde had a treatment scale of 100,000 $m^3$/d and a sewage-absorbing area of 38.14 $km^2$. The treated water was discharged into the eastern water system and the ecological treatment and then into the Majiaji River. Another one, namely, the Huangmuguan wastewater

treatment plant, lay in the southeastern district of the city and had a treatment capacity of 50,000 m³/d. The wastewater from the following five districts—the moat collection system, Nanzhu mountain sewage system, Tang Jiarong sewage collection system, BaiYuan pump station system and the Xinpo sewage pumping station system—was collected and discharged into this plant to be treated. After processing and meeting the emissions standards, the treated wastewater was discharged into Yuan River.

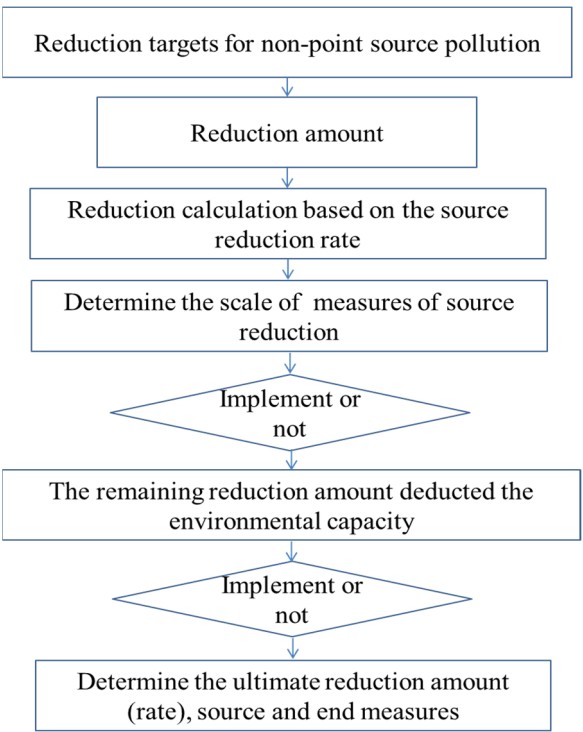

**Figure 10.** Technical route taken for pollutants reduction.

The non-point source pollution can be controlled by the source and end reduction measures. Source reduction measures mainly refer to the engineering and non-engineering measures that reduce the total amount of pollutants entering the radial flow. Specifically, it includes green roofs, rainwater tanks, permeable pavements, vegetation filter belts, planting grass ditches, seepage ditches, sand filter tanks and biological retention ponds. Through the rational allocation of source abatement measures, runoff pollution can be effectively reduced by 43%. The ecological filters constructed in the ship terminal area were the water treatment facility in operation. The non-point source pollutants could be reduced by 24% by the ecological filters based on the model calculation and the remaining 33% of the non-point source pollutants could be reduced by the end planning rainwater wetlands. According to the calculation, managing the remaining 33% of the pollutant needs endmost wetlands of 26.04 ha to be constructed. The area of the water bodies in Whitehorse Lake, Quyuan Park and Chuanzi River reach 43.39 ha. Therefore, wetlands construction is more feasible.

### 4.2.3. Water Ecosystem

The concept of low-impact development and ecology was accepted to protect the original ecologically sensitive water areas, such as rivers, lakes and wetlands, to maintain the natural hydrological characteristics before urban development. Meanwhile, the proportion of the impervious urban area should be controlled to reduce the damage of urban development to the original water ecological environment. In addition, ecological means are used to restore the water bodies and other natural environments that were damaged under the traditional urban construction mode.

Water ecological engineering can be divided into two parts: (1) runoff control engineering and (2) water ecological protection and restoration engineering. The latter includes comprehensive regulations of rivers and lakes and conservation projects for soil and water. Runoff control engineering can reach the goal of 78% annual runoff control based on low-impact development and the combination of basic gray and green measures. Furthermore, the comprehensive renovation projects of the rivers and lakes achieved the control target of a 90% ecological shoreline rate by transforming the hardened revetment of the river and lake system within the scope of special planning or protecting the original natural river and lake system.

By implementing low-impact development measures, different planning schemes and design guidelines are put forward for various plots and roads. In the process of site development, source and distributed rainwater runoff control measures were adopted to achieve runoff control objectives, such as ecological filter tanks, green roofs, biological retention zones and stepped rain gardens. The index decomposition of the planning area was divided into two levels. The first level was the decomposition of the planning area to the control unit and the second level was the decomposition of the control unit to the project. The specific index decomposition technology roadmaps of the two levels are shown in Figures 11 and 12.

The water bodies, shorelines and waterfront areas in the planning scope should be protected as a whole project, including water protection, water ecological protection, water quality protection and waterfront space control. Within the water control line, filling and occupying are not allowed and the integrity of the water body is maintained. The transformation of water bodies should be fully demonstrated. The construction shall ensure that the area of the blue line cannot be reduced. According to the Municipal Water System Planning Code, the protection scope of water areas requires delimiting.

Changde has numerous water systems. The city's internal water system can be closer to the people based on the construction of state revetment and a waterfront space with ecological beauty can be created. According to the ecological nature design concept, a river with an internal vertical stiffened revetment shorelines of streams and drainage channels was reconstructed to ensure the safety of stormwater, the river ecology and the landscape function.

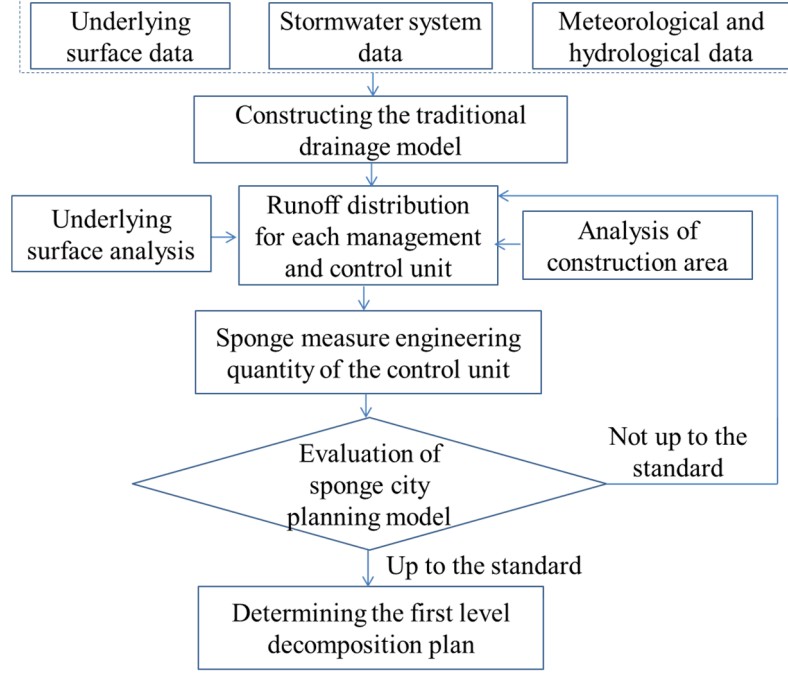

**Figure 11.** The first-order technology roadmap of the indicator decomposition.

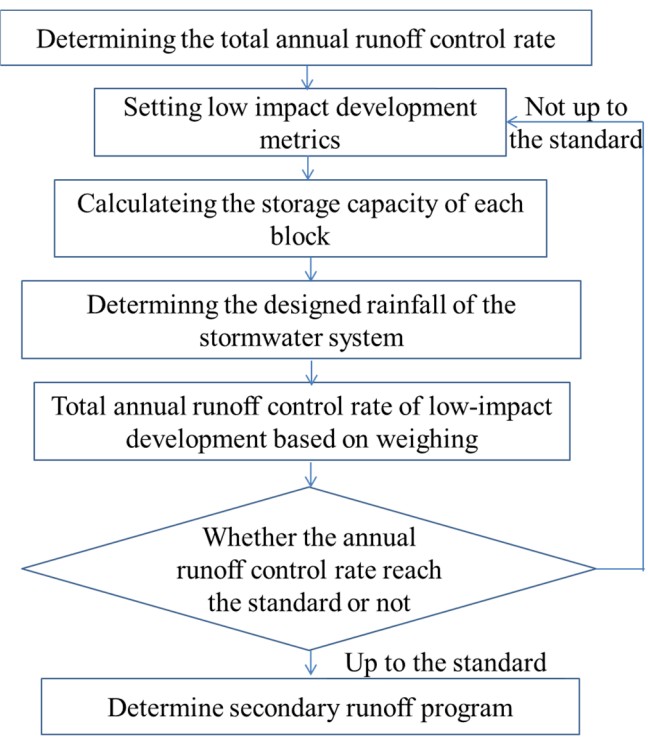

**Figure 12.** The second-order technology roadmap of the indicator decomposition.

### 4.2.4. Water Safety System

The water security system is mainly composed of the drainage and waterlogging prevention system and the flood control system. Furthermore, the drainage and waterlogging prevention system also includes a minor drainage system and a major drainage system. The technological roadmap of the water security system is presented in Figure 13.

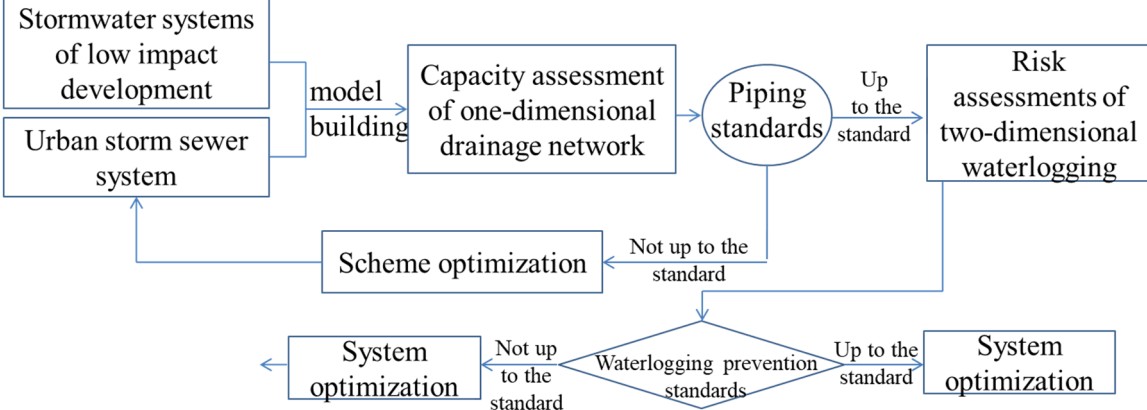

**Figure 13.** Technological roadmap of the water security system.

In light of the reconstruction of the drainage pipes in the old city, the new area was designed according to the shunt system to reach the goal of a complete shunt system in the drainage system in the area. According to the Special Drainage (Rainwater) Waterlogging Planning of Changde, the downtown area of Changde was divided into three high-discharge areas and thirty-nine areas. Combined with the geological conditions, road network planning, the built and planned facilities, cost, maintenance and management, and other factors, a reasonable layout of the urban drainage system was achieved. In view of the serious waterlogging situation in the Jiangbei urban area, the drainage capacity of the existing rainwater pumping station was evaluated and the water level of the inland river

system was analyzed. Moreover, a new urban drainage port with a higher design standard and a larger drainage capacity was built. The present rain-pumping station in the area is shown in Table 2. Considering the scale of cluster construction land, the flow direction of rainwater and the catchment area, Dongting Avenue in Jiangbei city was determined to be the main discharge channel of an excessive rainstorm.

**Table 2.** Pumping station in the planning area.

| Pumping Station | Catchment Area (ha) | Designed Drainage Capacity (m³/s) | Storage Volume (m³/s) | Planning Attribute |
|---|---|---|---|---|
| Chuanmatou station | 446.87 | 12.6 | 20,000 | Built |
| Gandang station | 138.25 | 9.56 | 5000 | New project |
| Changgang station | 273.16 | 13.52 | 9000 | New project |

Rainwater storage ponds and tanks were set beside the buildings and in the communities; wet ponds were built in the centralized green spaces to strengthen the landscape water storage functions so they could regulate and store collected rainwater for greening watering, road cleaning or landscape water replenishment. The space storage, such as the Whitehorse Lake and Chuanzi River, was fully utilized for storing the rain and flooding. Moreover, the stored rain can complement the urban water reuse system for the landscape and ecological water, among other aspects. Based on this, five percent of the utilization of rainwater resources can be realized.

In addition to the water loss, such as water surface evaporation, the annual rainwater collected in the area could fully meet the requirements of the regional greening and road needs and an additional supply of municipal water was not needed. The rainwater collected in February, September and December could not meet the total water consumption of the month. The surplus water could be collected for insufficient months. Rainwater barrels and storage ponds were used to regulate, store and utilize the rainwater in the buildings and residential plots, and a total of $0.24 \times 10^4$ m³ rainwater barrels and storage ponds were set up.

*4.3. Discussion*

Sponge city facilities greatly improved the waterlogging and urban water environment problems in the planned area. The water quality standard of surface water is now better than class IV, and the annual SS removal rate is over 45%, which was completed by improving the sewerage pipe network, accelerating the construction of sewage treatment plants and eliminating the direct sewage discharge. Taking the total nitrogen and total phosphorus concentrations as the research aims, Xiong et al. simulated the impact of Shenzhen sponge city construction on the water quality of the bay and concluded that the construction of a sponge city could also reduce the total nitrogen and total phosphorus concentrations in the bay by 2.4% and 20.2%, respectively [27]. The runoff pollution could be effectively reduced by 43% through the rational allocation of source abatement measures, such as planting grass ditches, seepage ditches, sand filter tanks and biological retention ponds. The non-point source pollutants could be reduced by 24% by the ecological filters and the remaining 33% of the pollutant need to be managed by constructing endmost wetlands with an area of 26.04 ha. The runoff control engineering can reach the goal of 78% annual runoff control based on low-impact development and the combination of basic gray and green measures. By strengthening rainwater collection and landscape water regulation and storage function, the target of 5% rainwater resource utilization was achieved. The rain barrel and rainwater storage tank totaled 0.24 million m³. The artificial wetlands (Whitehorse Lake, Quyuan Park Water, Chuanzi River) were created for rainwater regulation, storage and utilization, and the total regulation and storage capacity was 13.02 m³. After the construction of the sponge city, Ji et al. simulated the flood characteristics of the Fenghe River in Xixian New Area. The results implied that constructing the

sponge city could reduce the peak flow of urban inland river floods to a certain extent, which demonstrated that the sponge city construction positively affects the water quality and flood characteristics of downstream rivers [28]. Thus, we can conclude that sponge city construction plays an important role in reducing peak flood discharge, reducing runoff pollution, controlling urban waterlogging and improving the urban water environment.

The construction of the sponge city could reduce the flood loss of USD 16.5 million and save the drinking water cost of USD 1.07 million. The added land value benefit reached USD 440 million along the Chuanzi River and the moat. More than 1000 related jobs are expected to be created. The research results from Shen et al. showed that sponge city construction increased housing prices in Suining City (Sichuan, China) by 385 RMB/m$^2$, or 5.48%. If the reference value is a real estate price increase of 385 RMB/m$^2$, the benefits of constructing Xi'an and Guyuan's sponge cities are USD 6.4 billion and 5.3 billion, respectively [29]. It can be seen that sponge city constructions play a promoting role in improving regional land value.

The majority of citizens are significantly more satisfied with the government and urban environmental conditions, forming a strong sense of pride and belonging and strengthening social cohesion. Obviously, sponge city constructions have an influence on society and are reflections of the residents.

## 5. Conclusions

(1) Ninety-one sponge transformation projects of the old construction zone and the four projects of road and park sponging in Changde were completed to realize source pollution reduction. A total of 7.67 km of rainwater pipelines were newly constructed for process control. Regarding the end of pipe control, two park and green space sponging transformation projects were completed. Furthermore, eight water system regulation and ecological restoration projects and four pumping station and storage tank projects were completed.

(2) Taking the measured rainfall in 2019 as the simulation condition, the results showed that the annual net flow total control rate and the runoff pollution reduction reached 77.56% and 45.18%, respectively, after the transformation based on the sponge city concept, which met the target demand. The total runoff and peak flow were reduced by 35.08% and 26.82%, respectively. Meanwhile, the peak flow of the runoff pollution concentration was reduced by 31.99%. Therefore, the water quantity and quality targets could meet the assessment requirements based on the transformation of the sponge city in Changde.

(3) Through the construction of the sponge city, surface runoff pollutants were greatly reduced, and the pollutant load reduction rate of non-point source pollution in the area reached more than 45%. Thirty-eight kilometers of rivers and ditches were renovated. The river and lake ecosystems, water environment and the capacity of the urban water environment were greatly improved to achieve the unique natural landscape of green mountains, clear water, rivers, lakes and mountains. The project not only provided a reference for the planning and construction of a sponge city for other cities in China and presented the 'Standard Atlas of Sponge City' but also alleviated the problems of urban waterlogging and black and odorous water bodies and ensured sustainable development of urban water environment.

(4) Sponge city construction is a systematic project that requires the systematic deployment of 'source', 'process' and 'end' rainwater reduction control. Sponge construction cannot be used for sponge's sake, and the sponge transformation in built-up areas should be problem-oriented to effectively solve the current problems. The application of the drainage model should be given great importance to analyze and quantify the current problems and to complete the simulation assessment of the hydrologic and water quality compliance.

**Author Contributions:** Conceptualization, supervision, data collection and curation, Y.D. and J.D.; funding acquisition, review, writing and editing, C.Z. All authors read and agreed to the published version of the manuscript.

**Funding:** This work was financially funded by the Natural Science Foundation of Hunan Province (No. 2021JJ30080).

**Institutional Review Board Statement:** Not applicable.

**Informed Consent Statement:** Not applicable.

**Data Availability Statement:** Not applicable.

**Acknowledgments:** The authors sincerely appreciate all survey participants, the paper reviewers and "the Natural Science Foundation of Hunan Province, grant No. 2021JJ30080".

**Conflicts of Interest:** The authors declare no conflict of interest.

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
