# Peer review of "Sponge City and Water Environment Planning and Construction in Jibu District in Changde City"

_sustainability, doi:10.3390/su15010444_

Round 1
Reviewer 1 Report
This paper studies the current urban situation, rainfall type and water environment in the sponge construction area were analyzed and the reasons resulted urban waterlogging and deterioration of urban inland river water quality.I have some comments.
Author states that "In the conference of Low-carbon Cities and Regional Development Technology Forum held in 2012.04, the Sponge City concept was first proposed by the Ministry of Housing and Urban-Rural Development of China". When referring to a conference, historical event, or point in time, the author needs to give the source of the data, the relevant evidence.
The abstract should directly address your point of view.
The research contributions need to be described in Section 1.
The water body in the planning area was seriously polluted. The drainage system of the moat in the old city in the planning area was the intercepting combined confluence. It was constructed in the 1960s and 1970s. So how should the author address this issue in the latter way?
Author Response
Thank you for your work on the manuscript.
This paper studies the current urban situation, rainfall type and water environment in the sponge construction area were analyzed and the reasons resulted urban waterlogging and deterioration of urban inland river water quality. I have some comments.
1>: Author states that "In the conference of Low-carbon Cities and Regional Development Technology Forum held in 2012.04, the Sponge City concept was first proposed by the Ministry of Housing and Urban-Rural Development of China". When referring to a conference, historical event, or point in time, the author needs to give the source of the data, the relevant evidence.
Response: The information is available in China's official media, as well as in some relevant published papers. The relevant published references were added.
The following section is the revised part and has been added in the revised manuscript.
In the conference of Low-carbon Cities and Regional Development Technology Fo-rum held in 2012.04, the Sponge City concept was first proposed by the Ministry of Housing and Urban-Rural Development of China [1]. In addition, The Technical Guide for Sponge City Construction ‘Construction of Rainwater System for Low Impact De-velopment (Trial)’, was promulgated and implemented on October 22, 2014, which provided guidance and basis for the construction of Sponge City [2, 3].
2>: The abstract should directly address your point of view.
Response: The abstract has been revised according to your and the other reviewer’s advice.
The following paragraph is the revised abstract.
Abstract: Urban waterlogging and urban water environment problems in Changde city caused by extreme weather have seriously hindered the sustainable development of cities. Sponge City is not only the inheritance and development of foreign technology, but also a new way. The background of the Sponge City construction based on green infrastructures of in China was introduced in this paper. As one of the first pilot construction city based on sponge concept, Changde city possesses natural geographical advantages. The current urban situation, rainfall type and water environment in the sponge construction area were analyzed and the reasons resulted urban waterlogging and deterioration of urban inland river water quality were presented. On account of the urban water environment and ecological status, the specific strategic objectives of the Sponge City transfor-mation were given. Meanwhile, the overall technical route and the concrete realization path of each single index, such as water environmental system, water ecological system and security system were also presented. The annual net flow total control rate and the runoff pollution reduction reached 77.56% and 45.18%, respectively. The total runoff and peak flow were also reduced by 35.08% and 26.82%, respectively. Meanwhile, the peak flow of runoff pollution concentration was reduced by 31.99%. The pollutant load reduction rate of non-point source pollution in the area reached more than 45%. The project not only improved the problems of urban waterlogging and black and odorous water body, but also ensured the sustainable development of urban water environment.
3>: The research contributions need to be described in Section 1.
Response: The last paragraph in section 1 was revised according to the advice and published papers in this journal.
The following section is the revised part and has been added in the revised manuscript.
Summarily, the other parts of the paper are organized as follows: section 2 reviews the survival literature on the application of the advanced urban rainwater utilization system in different countries. Section 3 highlights the methodology and data used in the research. It presents the theoretical framework and the empirical model used in this attempt. The results and discussion are presented in the section 4. It presents the planning and design process of Sponge City and its operation as well as the improvement of urban water environment of the city. The conclusions are provided in Section 6. As one of the first Sponge City pilot construction cities in China, it can provide reference for later Sponge City planning and design of other cities.
4>: The water body in the planning area was seriously polluted. The drainage system of the moat in the old city in the planning area was the intercepting combined confluence. It was constructed in the 1960s and 1970s. So how should the author address this issue in the latter way?
Response: In the process of sponge city construction, the early construction of the confluence system in the area were transformed into the interception confluence system, and then into the completely separate drainage system.
Reviewer 2 Report
Overall, the manuscript is well written. However, the authors need to have following revisions:
1. To more clearly point out the research gaps of existing studies in the end of literature review.
2. To clearly present the theoretical contributions of the study in the conclusion section.
3. To indicate the practical implications of the study in the conclustion section.
Author Response
Thank you for your work on the manuscript.
Overall, the manuscript is well written. However, the authors need to have following revisions:
1>: To more clearly point out the research gaps of existing studies in the end of literature review.
Response: The relative contents have been added in the last paragraph in the section of literature review section.
The following section is the revised part and has been added in the revised manuscript.
Vast majority of cities in developed countries have gone through multiple stages from water supply city to water cycle city. The LID in America, SuDS in Britain and WSUD in Australia has experienced long-term development at different stages, the proposal of low-impact development focuses more on the control of small and medium rainfall events and takes the total runoff as the control target, and adopts the measures of source, dispersion and small green technology. However, the future development of Chinese cities does not necessarily need to follow the old roadmap and start from the drainage city again to reach the future Sponge City through the waterways city and water cycle city. Most cities in China are now in the primary stage of the waterways city; the urban development is still active and a large number of regional rectification and development projects still are under planning [25, 26]. Sponge City construction aims to transform the concept of urban development and build a sustainable urban development mode. Its core concept is consistent with American low-impact development. However, the low impact development technology measures in Sponge City construction include not only the green facilities at the source, but also the green rainwater infrastructure at different links and scales in the middle and end. Control targets should include comprehensive targets such as total runoff, peak value, pollution and frequency. Although the original intention of Sponge City construction is related to the solution of urban waterlogging, it should return to the premise of how to improve the sustainability of urban construction and urban livability in the final analysis, which also coincides with the connotation of water-sensitive city.
2>: To clearly present the theoretical contributions of the study in the conclusion section.
Response: The relative contents have been added in the section of conclusion and sighed with red font.
The following section is the revised part and has been added in the revised manuscript.
(3) Through the construction of Sponge City, surface runoff pollutants are greatly reduced, and the pollutant load reduction rate of non-point source pollution in the area reached more than 45%. Thirty-eight kilometers of rivers and ditches have been renovated. The river and lake ecosystems, water environment and the capacity of the urban water environment were greatly improved to achieve the unique natural landscape of green mountains, clear water, rivers, lakes and mountains. The project not only provided the reference of the planning and construction of Sponge City for the other cities in China and presented the ‘Standard Atlas of Sponge City’, but also improved the problems of urban waterlogging and black and odorous water body and ensured the sustainable development of urban water environment.
3>: To indicate the practical implications of the study in the conclustion section.
Response: The relative contents have been added in the section of conclusion and sighed with red font.
The following section is the revised part and has been added in the revised manuscript.
(3) Through the construction of Sponge City, surface runoff pollutants are greatly reduced, and the pollutant load reduction rate of non-point source pollution in the area reached more than 45%. Thirty-eight kilometers of rivers and ditches have been renovated. The river and lake ecosystems, water environment and the capacity of the urban water environment were greatly improved to achieve the unique natural landscape of green mountains, clear water, rivers, lakes and mountains. The project not only provided the reference of the planning and construction of Sponge City for the other cities in China and presented the ‘Standard Atlas of Sponge City’, but also improved the problems of urban waterlogging and black and odorous water body and ensured the sustainable development of urban water environment.
Reviewer 3 Report
Manuscript ID: sustainability-2076943-peer-review-v1
Title: Sponge City and water environment planning and Construction in Jibu district in Changde City
Journal: Sustainability
The article addresses a topic of interest and relevance, but presents some methodological problems. The paper structure and content should be highly improved for a scientific publication:
Ø The abstract is too generic. The abstract should consist of an easy to understand summary of the study, its methodology and obtained results. Please give the numerical results in this section. It should answer the following questions: What problem did you study and why is it important? What methods did you use? What were your main results? And what conclusions can you draw from your results?
Ø Intro The objectives of the study were defined. But, It is difficult to understand the research questions and what is expected from the study. How do these studies set up the context for this? Given the number of studies in this research area; what gap will this study fill in the literature? For example, what is the contribution/novelty of this study? What kind of local and global knowledge the authors want to improve?
Ø Meto: Materials and Methods are descriptive and provide any information related to the input data and the methodology used in the study. These should be added in detail to the method section.
The discussion section is missing. Regarding the discussion, the authors should further compare their findings with references. As you included a very wide range of background information, a more structured illustration of these background literature references could be promoted to add another benefit to the paper. Link your findings with those from previous studies and this will also help make more broader conclusions. The results should be further elaborated to show how they could be used for the real applications in other similar climate regions. Discussion of the results has never been done and needs to be critically analyzed. The discussion section is very important for the results of the study. Please use relevant new references: DOI 10.1016/j.uclim.2021.101058; DOI 10.3390/su141811653; DOI 10.20937/ATM.2017.30.04.06; DOI 10.1007/s11356-022-22297-1; DOI 10.1007/s11356-020-10555-z ….etc. All data should be analyzed statistically.

Author Response
Thank you for your work on the manuscript.
The article addresses a topic of interest and relevance, but presents some methodological problems. The paper structure and content should be highly improved for a scientific publication:
1>: The abstract is too generic. The abstract should consist of an easy to understand summary of the study, its methodology and obtained results. Please give the numerical results in this section. It should answer the following questions: What problem did you study and why is it important? What methods did you use? What were your main results? And what conclusions can you draw from your results?
Response: The abstract has been checked and revised according to the advice. Relative contents have been added.
The following paragraph is the revised abstract.
Abstract: Urban waterlogging and urban water environment problems in Changde city caused by extreme weather have seriously hindered the sustainable development of cities. Sponge City is not only the inheritance and development of foreign technology, but also a new way. The background of the Sponge City construction based on green infrastructures of in China was introduced in this paper. As one of the first pilot construction city based on sponge concept, Changde city possesses natural geographical advantages. The current urban situation, rainfall type and water environment in the sponge construction area were analyzed and the reasons resulted urban waterlogging and deterioration of urban inland river water quality were presented. On account of the urban water environment and ecological status, the specific strategic objectives of the Sponge City transfor-mation were given. Meanwhile, the overall technical route and the concrete realization path of each single index, such as water environmental system, water ecological system and security system were also presented. The annual net flow total control rate and the runoff pollution reduction reached 77.56% and 45.18%, respectively. The total runoff and peak flow were also reduced by 35.08% and 26.82%, respectively. Meanwhile, the peak flow of runoff pollution concentration was reduced by 31.99%. The pollutant load reduction rate of non-point source pollution in the area reached more than 45%. The project not only improved the problems of urban waterlogging and black and odorous water body, but also ensured the sustainable development of urban water environment.
2>: Intro The objectives of the study were defined. But, It is difficult to understand the research questions and what is expected from the study. How do these studies set up the context for this? Given the number of studies in this research area; what gap will this study fill in the literature? For example, what is the contribution/novelty of this study? What kind of local and global knowledge the authors want to improve?
Response: The study aimed to solve the urban waterlogging and urban water environment problems in the city and provided empirical reference of the planning and construction of sponge city for other cities. Due to differences in natural topography and urban water system, different cities may adopt different technologies and methods in sponge city planning, design and construction, but the overall guiding ideology may be similar. The core concept of sponge city in China is consistent with the low-impact development in the United States. However, due to the long-term weak development of urban rainwater in China, multiple problems such as urban waterlogging and runoff pollution have been accumulated. The technical measures not only include the green facilities at the source, but also cover the green rainwater infrastructure at different links and scales in the middle and end. The control targets include comprehensive targets such as total runoff, peak value, pollution and frequency, etc., which can be applied to each subsystem of sponge city technology according to actual conditions. The study can provide the personalized design case of sponge city. Relative contents have been added in the section 4.1.
4.1. Objective and Strategy
According to the concept of ‘water saving priority, spatial balance and systematic governance’, a Sponge City construction framework that conformed to the natural environment characteristics of the airport area and the actual urban development were put forward. The planning ideas aimed to lead urban development with the Sponge City construction concept, to promote the ecological protection, economic and social development and cultural inheritance, and to build a new image of Changde with ecological, safe and dynamic sponge construction and then to realize the development strategy of ‘water security guarantee, water environment improvement, good water ecology, beautiful water landscape’. The study aimed to solve the urban waterlogging and urban water environment problems in the city and provided empirical reference of the planning and construction of Sponge City for other cities with the personalized design case.
3>: Meto: Materials and Methods are descriptive and provide any information related to the input data and the methodology used in the study. These should be added in detail to the method section. The discussion section is missing. Regarding the discussion, the authors should further compare their findings with references. As you included a very wide range of background information, a more structured illustration of these background literature references could be promoted to add another benefit to the paper. Link your findings with those from previous studies and this will also help make more broader conclusions. The results should be further elaborated to show how they could be used for the real applications in other similar climate regions. Discussion of the results has never been done and needs to be critically analyzed. The discussion section is very important for the results of the study.
Response: Because the study belongs to planning and design article, not an experimental research article, so the material part of the paper is not designed. The relative basic data for the sponge city design, such as the area of the planned plot, area and proportion of underlying surface, elevation, slope, groundwater level, formula of rainstorm intensity, the rainfall type and relevant basic data of urban water environment etc are presented in the section of methodology.
Your comments about the comparison of references are very pertinent. In our previous articles, we also attach great importance to the comparative analysis of experimental results with existing literature. However, such engineering design articles may have difference with the scientific research papers. Due to the different basic conditions, there may be different personalized design schemes. The papers presented by the reviewer (DOI 10.1016/j.uclim.2021. 101058; DOI 10.3390/su141811653) are also in the field of sponge city and in the section of discussion, there is description of the design results and no direct comparison.
Due to the different geographical and hydrological conditions of each city, the sponge city design theory, method and operation results in this city can only provide reference. Just like the rainstorm intensity formula, it can be obtained according to the annual rainfall, but the calculation method of the net rainstorm flow in the process of sponge city design can be similar.
According to the reviewer’s advice, the section of discussion was added. The References were also added.
The following paragraphs are the added discussion part.
4.3 Discussion
The sponge city facilities have greatly improved the waterlogging and urban water environment problems in the planned area. The water quality standard of surface water now is better than class IV, and the annual SS removal rate is over 45%, which was completed by improving sewage pipe network, accelerating the construction of sewage treatment plants and eliminating the direct sewage discharge. Taking as the total nitrogen and total phosphorus concentrations as the research aims, Xiong et al simulated the impact of Shenzhen Sponge City construction on the water quality of the bay, and concluded that the construction of Sponge City can also reduce the total nitrogen and total phosphorus concentrations in the bay by 2.4% and 20.2%, respectively[24]. The runoff pollution can be effectively reduced by 43% through rational allocation of source abatement measures, such as planting grass ditch, seepage ditch, sand filter tank and biological retention pond ect. The non-point source pollutants can be reduced by 24% by the ecological filters and the remaining 33% of the pollutant needs constructed endmost wetlands of 26.04ha. The runoff control engineering can reach the goal of 78% annual runoff control based on the low-impact development and the combination of basic gray and green measures. By strengthening rainwater collection and landscape water regulation and storage function, the target of 5% rainwater resource utilization is achieved. The rain barrel and rain water storage tank total 0.24 million m3. The artificial wetlands (Whitehorse Lake, Quyuan Park Water, Chuanzi River) for rainwater regulation, storage and utilization, and the total regulation and storage capacity is 13.02 m3. After the construction of Sponge City, Ji et al simulated the flood characteristics of the Fenghe River in Xixian New Area. The results implies that constructing the Sponge City could reduce the peak flow of urban inland river floods to certain extent, which demonstrates that the Sponge City construction positively affects the water quality and flood characteristics of downstream rivers[25]. Thus, we can conclude that Sponge City construction plays an important role in reducing peak flood discharge, reducing runoff pollution, controlling urban waterlogging and improving urban water environment.
The construction of sponge city can reduce the flood loss of 16.5 million $ and save the drinking water cost of 1.07 million $. The added land value benefit reached 440 million $ along the Chuanzi river and the moat. And more than 1,000 related jobs are expected to be created. The research results from Shen et al show that Sponge City construction increased housing prices in Suining City (Sichuan, China) by 385 $/m2, which is 5.48%. If the reference value is a real estate price increase of 385 $/m2, the benefits of constructing the Xi’an and Guyuan’s Sponge Cities are 6.4 and 5.3 billion RMB, respectively [26]. It can be seen that sponge city construction plays a promoting role in improving regional land value.
The majority of citizens are significantly more satisfied with the government and urban environmental conditions, forming a strong sense of pride and belonging, and strengthening social cohesion. Obviously, the construction of the sponge cites observe its influence on society and survey the reflections of the residents.
Round 2
Reviewer 3 Report
The present work could be interesting for the future urban planning for suistanable cities. I appreciate and understand the quality of your journal and the steadfast guidance to have clearly expressed ideas of the research to be published. At the present version of this manuscript I believe that it now are able express that this research is comprehensive for these type of areas and the results can easily be extended to regions of similar climate.
At this point, the paper have revisions restructuring. Therefore, I can recommend publication of the current version.
I recommend that this paper could be accepted for this journal.